# Node Embeddings and Exact Low-Rank Representations of Complex Networks

**Sudhanshu Chanpuriya**
University of Massachusetts Amherst
schanpuriya@umass.edu

**Cameron Musco**
University of Massachusetts Amherst
cmusco@cs.umass.edu

**Charalampos E. Tsourakakis**
Boston University & ISI Foundation
tsourolampis@gmail.com

**Konstantinos Sotiropoulos**
Boston University
ksotirop@bu.edu

## Abstract

Low-dimensional embeddings, from classical spectral embeddings to modern neural-net-inspired methods, are a cornerstone in the modeling and analysis of complex networks. Recent work by Seshadhri et al. (PNAS 2020) suggests that such embeddings cannot capture local structure arising in complex networks. In particular, they show that any network generated from a natural low-dimensional model cannot be both sparse and have high triangle density (high clustering coefficient), two hallmark properties of many real-world networks.

In this work we show that the results of Seshadhri et al. are intimately connected to the model they use rather than the low-dimensional structure of complex networks. Specifically, we prove that a minor relaxation of their model can generate sparse graphs with high triangle density. Surprisingly, we show that this same model leads to *exact low-dimensional factorizations* of many real-world networks. We give a simple algorithm based on logistic principal component analysis (LPCA) that succeeds in finding such exact embeddings. Finally, we perform a large number of experiments that verify the ability of very low-dimensional embeddings to capture local structure in real-world networks.

## 1 Introduction

Graphs naturally model a wide variety of complex systems including the internet, social networks, transportation networks, protein-protein interaction networks, the human brain, and co-authorship networks. Understanding and analyzing such networks lies at the heart of computer science. In recent years there has been a surge of interest in developing node embedding techniques that map the nodes of a graph to low-dimensional Euclidean space in such way that the geometry of the embedding reflects important structure in the graph. Specifically, a node embedding method takes as input a graph $G$ with $n$ nodes $v_1, \ldots, v_n$ and maps each node $v_i$ to a vector $x_i \in \mathbb{R}^k$, where $k$ is an embedding dimension typically with $k \ll n$. The learned embeddings can be used as input for downstream machine learning tasks such as clustering, classification, and link prediction [HYL17].

Geometric representations of graphs and low-rank factorizations have a long history, cf. the text of Lovász and Vesztergombi [LV99], and important successes including spectral clustering [SM00, NJW02], Laplacian eigenmaps [BN03], IsoMap [TDSL00], locally linear embeddings [RS00], and community detection algorithms [ABH15, McS01, RCY+11, CRV15]. The stunning successes of deep learning in recent years have also led to a new generation of neural network-based node embedding methods. Such methods include DeepWalk [PARS14], node2vec [GL16], LINE [TQW+15], NetMF [QDM+18], and many others [TQM15, CLX16, KW16, WCZ16].

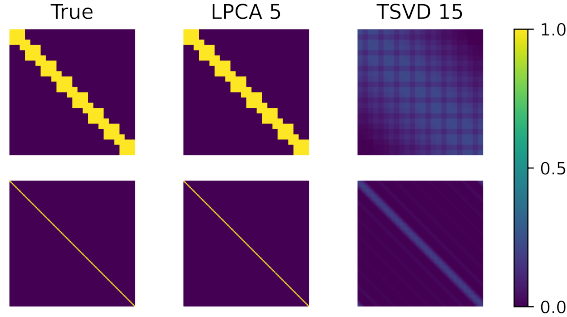

Figure 1: Reconstructions of a toy graph with 100 triangles connected in a loop and a self-loops on each vertex. Top: Zoomed in to the first 24 vertices, i.e., the first 8 triangles; Bottom: Whole graph. Left: True adjacency matrix; Middle: rank-5 approximation produced by our logistic PCA variant (LPCA); Right: rank-15 approximation with truncated SVD (TSVD) method [SSSG20].

This recent explosion of novel node embedding methods has already proved valuable for numerous graph mining tasks. But what are their limitations? This question was recently posed by Seshadri et al. in their PNAS paper [SSSG20]. Seshadri et al. remark that (i) regardless of the node embedding method, the goal is to produce a low-dimensional embedding that captures as much structure in $G$ as possible, and that (ii) it is well-known that real-world networks are sparse in edges, and rich in triangles. They ask an intriguing question: can low-dimensional node embeddings represent triangle-rich complex networks? Their key conclusion is that graphs generated from low-dimensional embeddings cannot contain many triangles on low-degree vertices, and thus the answer to the aforementioned question is negative. See Theorem 4 in Section 2 for a formal statement of this result.

In this work, we prove that the results in [SSSG20] are a consequence of the model they use, rather than a general property of low-dimensional embeddings. We state our contributions as informal results; for the formal statements, see Section 3. Our first main result is:

> **Result 1.** Low-dimensional node embeddings are able to represent triangle-rich graphs.

Figure 1 give an illustrative example of Result 1. Consider the family of graphs consisting of a set of triangles connected in a cycle. This family is a hard instance according to result of [SSSG20] since it has near maximum triangle density given its low maximum degree. Indeed, we can observe that an optimal 15-dimensional representation using the proposed method of [SSSG20] preserves very little structure in the graph. However, as Figure 1 shows, there exists a rank-5 representation which nearly fully captures the graph structure. We discuss the details of this toy experiment in Section 4.

Our second key result is a low-dimensional model that can perfectly capture *all bounded degree graphs*, regardless of structure. An important corollary of this result is that preferential attachment graphs [BA99] admit a $\Theta(\sqrt{n})$-rank factorization with high probability without losing any information about their structure. Furthermore, our result is constructive.

> **Result 2.** There exists a low-rank factorization algorithm that provably produces exact low-dimensional embeddings for bounded degree graphs.

We believe such *exact embeddings* are of independent interest to researchers working on the interplay between privacy and node embeddings, e.g., [ECS+19, ZCZ+20] and on graph autoencoders [TGC+14, WCZ16, PHL+18, SHV19].

We complement our results with several experiments on real-world networks. We observe that a simple algorithm can produce *very low-dimensional* exact representations, that go below the theoretical bounds we prove in Result 2, and still preserve all local structure. See Table 1 for a preview of our results on some popular datasets. We show that even lower dimensional factorizations, while not exact, suffice to capture important structure such as degree and triangle density.

> **Result 3.** Empirically we observe that our proposed algorithm produces *very low-dimensional* embeddings that preserve the local structure of large real-world networks.

| Dataset | # Nodes | Mean Degree | **Exact Factorization Dimension** |
|---|---|---|---|
| Pubmed | 19 581 | 4.48 | 48 |
| ca-HepPh | 11 204 | 21.0 | 32 |
| BlogCatalog | 10 312 | 64.8 | 128 |
| Citeseer | 3 327 | 2.74 | 16 |
| Cora | 2 708 | 3.90 | 16 |

Table 1: Preview of our results; real-world graphs admit *exact* low-rank factorizations. For details and results on more datasets, see Section 4.

## 2   Related Work

**Node embeddings.** Low-dimensional node embeddings have a long and rich history. They have played a major role in theoretical computer science, and specifically in the development of approximation algorithms for NP-hard problems Spectral clustering relies on node embeddings based on a small number of extremal eigenvectors of a matrix representation of the graph (e.g., top eigenvectors of the adjacency matrix [McS01], or bottom eigenvectors of the Laplacian [AM85]) to find good cuts, e.g., [NJW02, SM00, RCY$^+$11]. Metric embeddings have been used in multi-commodity flow algorithms [LR99, LLR95]. These methods aim to embed a graph in a Euclidean space so that the distances between nodes in the graph are close to the geometric distances between the embeddings. In particular, by Bourgain's theorem, every metric space with $n$ elements (including every $n$ node graph metric) can be embedded in an $O(\log n)$ dimensional space with $O(\log n)$ distortion [Bou85]. Goemans and Williamson inaugurated through their seminal work on MAX-CUT [GW95] a large family of semidefinite programming relaxations for NP-hard problems that embed nodes in a high-dimensional space, and then round these embeddings to obtain a near-optimal solution.

Many of the most classic random network models, including the stochastic block model [HLL83, ABFX08] and random dot-product models [YS07] are based on low-dimensional node embeddings. Each pair of nodes $v_i, v_j$ is connected with probability depending on the similarity between their embeddings (e.g., the dot product $x_i^T x_j$.). Many machine learning methods learn a latent low-dimensional embedding by maximizing a likelihood function that depends on a probabilistic model of this form [KTG$^+$06, MJG09, PKG12, GB13]. Relatedly, work in non-linear dimensionality reduction learns node embeddings that capture general dataset structure. Classical methods such as Laplacian eigenmaps, IsoMap, and locally linear embeddings [BN03, TDSL00, RS00] associate a graph $G$ with a generic high-dimensional dataset (e.g., by forming a $k$-nearest neighbors graph) and apply variants of spectral embedding on $G$ to find an informative embedding for the original data points.

In recent years there has been a surge of interest in node embedding methods inspired by the successes of deep learning [LBH15]. Neural network based methods including DeepWalk, Node2Vec, LINE, PTE, SDNE, and many more [PARS14, TQW$^+$15, TQM15, GL16, CLX15, WCZ16, CLX16, PHL$^+$18, WWW$^+$18] have become the node embeddings of choice in practice. Better understanding these methods is an active area of research. Some of them can be viewed as implicitly factoring a matrix corresponding to the graph $G$, connecting them to classic work on spectral embedding [LG14, QDM$^+$18, CM20]. Recently, Seshadri et al. [SSSG20] asked a crucial question: are there any inherent limitations on the ability of low-dimensional embeddings to capture relevant structure in complex networks? We discuss their paper next, as it is the key motivation behind our work.

**Impossibility of low-rank representations of triangle-rich networks [SSSG20].** Seshadri et al. argue that low-dimensional embeddings provably cannot capture important properties of real-world complex networks. In particular, it is well-known that real-world networks are sparse and contain many triangles, see e.g., [FMT09, LKF07]. They argue that a graph generated from a natural low-dimensional embedding cannot have this property. In particular they consider a truncated dot product model, where each node $v_i$ is associated with an embedding $x_i \in \mathbb{R}^k$ and nodes $v_i, v_j$ connect with probability proportional to the dot product $x_i^T x_j$, truncated to lie in $[0, 1]$. Formally:

**Theorem 4** (Theorem 1 of [SSSG20]). *Let $A = \sigma(XX^T)$ where $X \in \mathbb{R}^{n \times k}$ and $\sigma(x) = \max(0, \min(1, x))$ is a thresholding function which is applied entry-wise to $XX^T$. For any $c \geq 4$ and $\Delta \geq 0$, if a graph $G$ is generated by adding edge $(i, j)$ independently with probability $A_{i,j}$ and the expected number of triangles in $G$ that only involve vertices of expected degree $\leq c$ is $\geq \Delta n$, then the embedding dimension $k \geq \min(1, \Delta^4/c^9) \cdot n/\log^2 n$.*

If the triangle density $\Delta$ and the maximum degree $c$ are fixed, Theorem 4 implies that $X$ must have near-linear dimension $k = O(n/\log^2 n)$. That is, no low-dimensional embedding can capture the important feature of high triangle density on low-degree nodes. This result contrasts with the well-known fact that low-rank approximations can be used to approximate global triangle counts [Tso08], showing that counts restricted to a subgraph of bounded degree nodes cannot be preserved.

Seshadhri et al. conjecture that Theorem 4 generalizes to models where $A_{i,j}$ is generated by natural functions of the embeddings $x_i, x_j$ other than the truncated dot product. In the next section we argue that Theorem 4 is in fact brittle and a consequence of the specific matrix factorization model used.

## 3 Theoretical Results

We start by showing the impossibility result of Theorem 4 depends critically on the fact that each node is associated with just a single embedding $x_i \in \mathbb{R}^k$. This ensures that the low-rank matrix $XX^T$ is positive semidefinite (PSD), which is key in proving Theorem 4. Many network embeddings, such as DeepWalk and Node2Vec produce two embeddings $x_i, y_i \in \mathbb{R}^k$ for each node – sometimes called "word" and "context" embeddings due to their use in the word embedding literature [MSC+13]. This leads to a factorization of the form $XY^T$ for $X, Y \in \mathbb{R}^{n \times k}$ which is not necessarily PSD. Further, as discussed in [SSSG20], other methods [HRH02] base connection probability on the Euclidean distance between $k$-dimensional points. The underlying squared Euclidean distance matrix $D \in \mathbb{R}^{n \times n}$ is known to be exactly factorized as $D = XY^T$ for $X, Y \in \mathbb{R}^{d \times k+2}$.

We show that this simple relaxation to allow for a non-PSD factorization allows extremely low-dimensional embeddings to capture sparse, triangle dense graphs.

**Theorem 5** (Low-Dimensional Embeddings Capture Triangles). *Let $A = \sigma(XY^T)$ where $X, Y \in \mathbb{R}^{n \times 3}$ and $\sigma(x) = \max(0, \min(1, x))$ is applied entrywise to $XY^T$. For any integer $c > 0$, there exist $X, Y$ such that if a graph $G$ is generated by adding edge $(i, j)$ independently with probability $A_{i,j}$ then $G$ has maximum degree $< c$ and $G$ contains $\Omega(c^2 n)$ triangles.*

We note that $G$ generated in Theorem 5 is just a union of $\frac{n}{c}$ $c$-cliques and in fact has the maximum triangle density possible for a graph with max degree $c$. $G$'s adjacency matrix $A$ is block diagonal with blocks of size $c$. This matrix is very far from low-rank and cannot be well approximated by a low-rank factorization $XY^T$. However, as we will see, the simple $\sigma(\cdot)$ non-linearity is quite powerful here, allowing an exact factorization of the form $\sigma(XY^T)$ for $X, Y \in \mathbb{R}^{n \times 3}$.

*Proof of Theorem 5.* We show how to form $X, Y \in \mathbb{R}^{n \times 3}$ such that $A = \sigma(XY^T)$ is the adjacency matrix for a union of $n/c$ cliques. Each clique contains $\binom{c}{3}$ triangles. Thus, there are $\frac{n}{c} \cdot \binom{c}{3} = \Omega(c^2 n)$ triangles in the graph, with maximum degree $c - 1$, giving the theorem.

We place $n$ points along a line in $n/c$ clusters of $c$ nodes each. All points in a cluster are very close to each other, and clusters are spaced far apart. We then consider a constant matrix minus the squared distance matrix between these points. We can observe that this matrix has rank at most 3: consider $x \in \mathbb{R}^{n \times 1}$ which represents the $n$ positions on the line. Let $x_2$ contain the entry-wise squares of the values in $x$. Let $D = x_2 1^T + 1 x_2^T - 2xx^T$ be the matrix whose entries are the squared Euclidean distances between the points in $x$. Let $\bar{D} = 2J - D$, where $J$ is the all ones matrix. Note that $\bar{D}$ has rank $\leq \text{rank}(2J - x_2 1^T) + \text{rank}(-1 x_2^T) + \text{rank}(-2xx^T) = 3$ and so can be written as $XY^T$ for $X, Y \in \mathbb{R}^{n \times 3}$. We set $A = \sigma(XY^T) = \sigma(\bar{D})$.

Choose the points in $u$ such that the clusters are separated by distance $> 2$ and within each cluster the $c$ points are arbitrarily close. For $i, j$ in the same cluster, $A_{i,j} = \bar{D}_{i,j} = 2 - \|u_i - u_j\|_2^2 > 1$ and so we have an edge in $G$ with probability 1. For $i, j$ in different clusters, $A_{i,j} = \bar{D}_{i,j} = 2 - \|u_i - u_j\|_2^2 \leq 0$, and so they do not have an edge in $G$. Thus, $G$ consists of a union of $n/c$ disjoint $c$-cliques. $\square$

The embedding of Theorem 5 might seem unnatural. E.g., different nodes have embeddings of very different lengths depending on their position along the line. However, it is not hard to reconstruct a similar idea using simple binary embeddings. See Appendix A. Additionally, as we saw in Section 1 (see also Section 4) an accurate rank-5 factorization for a closely related triangle dense graph (with $n/3$ triangles connected in a cycle), can be found with a simple logistic PCA method.

**Exact Embeddings of Bounded-Degree Graphs.**   Observe that in the proof of Theorem 5 we use the thresholded dot product model of [SSSG20] in a very restricted way: all entries of $XY^T$ are either $> 1$ or $< 0$ and thus all large entries are thresholded to 1 in $\sigma(XY^T)$ and all small entries to 0. Thus, the same example would hold if we replaced $\sigma$ with the sign function $s$ with $s(x) = 0$ for $x < 0$ and $s(x) = 1$ otherwise. In other words, our example relies on the fact that the adjacency matrix of $G$ has low *sign-rank*. It can be written as $A = s(XY^T)$ for $X, Y \in \mathbb{R}^{n \times 3}$. The sign-rank is widely studied due to its connections to circuit complexity [RS10, BT16], communication complexity [AFR85, LMSS07], and learning theory [AMY16]. It is known via a polynomial interpolation argument [AFR85] that any matrix with sparse rows or columns has low sign-rank, depending linearly on the sparsity. This yields:

**Theorem 6** (Exact Embeddings for Bounded-Degree Graphs)**.** *Let $A \in \{0, 1\}^{n \times n}$ be the adjacency matrix of a graph $G$ with maximum degree $c$. Then there exist embeddings $X, Y \in \mathbb{R}^{n \times (2c+1)}$ such that $A = \sigma(XY^T)$ where $\sigma(x) = \max(0, \min(1, x))$ is applied entry-wise to $XY^T$.*

For completeness, we give a proof of Theorem 6 in Appendix A, following the approach of [AFR85]. Theorem 6 stands in sharp contrast to the impossibility result of [SSSG20] (Theorem 4). Not only can low-rank models capture complex network structure, but they can capture the structure of *any bounded-degree graph* with rank depending only on the max degree. We remark that the technique used to prove Theorem 6 applies also when each row of $A$ is block sparse – with a few contiguous blocks of ones. Considering the union of cliques example in Theorem 5, if we set the diagonal of $A$ to one, we have a block diagonal matrix – each row has a single contiguous block of $c$ ones. This matrix thus has sign rank at most $2 \cdot 1 + 1 = 3$, giving an alternative proof of Theorem 5.

An interesting corollary of Theorem 6 is that even *random graphs* admit exact low-dimensional factorizations if they have bounded degree. For example, preferential attachment graphs [BA99], which bear certain similarities with real-world networks, are sparse graphs with maximum degree bounded by $O(\sqrt{n})$ with high probability [BRST01, FFF05]. We thus have:

**Corollary 7.** *A random preferential attachment graph with $n$ nodes generated according to the Barabási-Albert-Bollobás-Riordan [BA99, BRST01] model admits an exact $\Theta(\sqrt{n})$ factorization.*

Corollary 7 applies to numerous other random graph models with power law degree distributions as long as the maximum degree produced is sublinear, e.g., [DEM01, BO04, FT12].

We can interpret Theorem 6 and Corollary 7 in multiple ways: they illustrate the power of low-dimensional models to exactly represent local structure in sparse graphs. At the same time, they show that the goal of finding a low-dimensional embedding to reconstruct a graph may be misleading, since a sufficiently optimized embedding can interpolate and maybe 'over-fit' any bounded degree graph. This emphasizes that obtaining low or even zero approximation error graph embedding may simply be due to capturing the fact that the given graph has low maximum degree.

We will see that in practice, the bound of Theorem 6 is not tight. Via a simple logistic PCA method, we can construct very low-dimensional exact factorizations of many real-world graphs, even when they have high max degree. The precise description of our algorithm follows in Section 4.

## 4   Empirical Results

We now empirically evaluate the effectiveness of low-dimensional embeddings in capturing graph structure, showing that a simple approach can find exact embeddings that match and in fact out perform our theoretical bounds. Code is available at `https://github.com/schariya/exact-embeddings`.

**Datasets.**   Our evaluations are based on 11 popular real-network datasets, detailed below. Table 2 lists and shows some statistics of these datasets. For all networks, we ignore weights (setting non-zero weights to 1) and remove self-loops where applicable.

>   **PROTEIN-PROTEIN INTERACTION (PPI)** [SBCA$^+$10] is a subgraph of the PPI network for Homo Sapiens. Vertices represent proteins and edges represent interactions.

>   **WIKIPEDIA** [GL16] is a co-occurrence network of words from a subset of the Wikipedia dump. Nodes represent words and edges represent co-occurrences within length 2 windows.

**BLOGCATALOG** [ALM+09] is a social network of bloggers. Edges represent friendships.

**FACEBOOK** [LM12] is a subset of the Facebook social network collected from survey participants.

**CA-HEPPH** and **CA-GRQC** [LKF07] are collaboration networks from the "High Energy Physics - Phenomenology" and "General Relativity and Quantum Cosmology" categories of arXiv, respectively. Nodes represent authors, and two authors are connected if they have coauthored a paper.

**PUBMED** [NLG+12] consists of scientific publications from the PubMed database pertaining to diabetes. Nodes are publications, and edges represent citations among them.

**P2P-GNUTELLA04** [LKF07] is a snapshot of the Gnutella peer-to-peer network from 8/4/2002. Nodes are hosts in Gnutella, and directed edges are connections between hosts.

**WIKI-VOTE** [LHK10] represents voting on Wikipedia till January 2008. In particular, nodes are users that either request adminship or vote for/against such a promotion. A directed edge from node $i$ to node $j$ represents that user $i$ voted on user $j$.

**CITESEER** [SNB+08] represents papers from six scientific categories as nodes and the citations among them as directed edges.

**CORA** [SNB+08] contains machine learning papers. Each node is a publication, and there is a directed edge from node $i$ to node $j$ when paper $i$ cites paper $j$.

**Reconstruction Algorithms.** The empirical results of Seshadhri et al. [SSSG20] focus on the Truncated SVD (TSVD) algorithm. Let $Z \in \mathbb{R}^{n \times k}$ be the orthonormal matrix whose columns comprise the eigenvectors of the adjacency matrix $A \in \{0,1\}^{n \times n}$ corresponding to the $k$ largest magnitude eigenvalues. Let $W \in \mathbb{R}^{k \times k}$ be diagonal, with entries corresponding to the top $k$ eigenvalues. The TSVD embeddings are given by $X = Zs(W)|W|^{1/2}$ and $Y = Z|W|^{1/2}$, where $s(\cdot)$ denotes the sign function and all functions are applied entry-wise to $W$. To form an expected adjacency matrix, we compute $\sigma(XY^T)$ where $\sigma(x) = \max(0, \min(1, x))$ is applied element-wise.

Note that $XY^T$ produced by TSVD is the rank-$k$ matrix that is closest to $A$ in terms of Frobenius norm. I.e, it would be an optimal low-rank factorization *if the threshold $\sigma(\cdot)$ were not applied*. As discussed in Section 3, many natural adjacency matrices, especially triangle dense ones such as the example of Theorem 5, are very far from low-rank and thus $XY^T$ does not well approximate $A$ either both before and after the threshold.

This motivates our proposed embedding method, which is based on Logistic PCA (LPCA). Rather than minimizing the error between $XY^T$ and $A$, we attempt to directly minimize the error between $\sigma(XY^T)$ and $A$. For efficiency, we replace $\sigma$ with a natural smooth surrogate: the logistic function (the sigmoid). Specifically, given $A \in \{0,1\}^{n \times n}$ and embeddings $X, Y \in \mathbb{R}^{n \times k}$, we let $\tilde{A} = 2A - 1$ denote the shifted adjacency matrix with $-1$'s in place of $0$'s and use the loss function:

$$L = \sum_{i=1}^{n} \sum_{j=1}^{n} -\log \ell \left( \tilde{A}_{i,j} [XY^T]_{i,j} \right), \tag{1}$$

where $\ell(x) = (1 + e^{-x})^{-1}$ is the logistic function. We initialize elements of the factors $X, Y$ independently and uniformly at random on $[-1, +1]$. We find factors that approximately minimize the loss using the SciPy [JOP+ ] implementation of the L-BFGS [LN89, ZBLN97] algorithm with default hyper-parameters and up to a maximum of 2000 iterations. We check for exact factorization by comparing $A$ to $\sigma(XY^T)$. If these are not equal, the factorization is inexact; in that case, to reconstruct an expected adjacency matrix, we apply the logistic function $\ell$ entry-wise to $XY^T$.

**Toy Graph.** We return to the initial demonstrative example from Section 1 where we considered the family of graphs consisting of a set of $t$ triangles connected in a cycle (Figure 1). This family is interesting as it has near maximum triangle density given its sparsity. It starkly illustrates the difference in the capacities of LPCA and TSVD. With embeddings of rank 5, our LPCA method reconstructs a graph with 100 triangles with only minor errors. By contrast, elements of the reconstruction from TSVD at rank 5 are too small to visualize effectively on the same scale, with a maximum below 0.08; even at rank 15, TSVD struggles to capture this graph, significantly diffusing the mass of the adjacency matrix away from the diagonal. In particular, the relative Frobenius errors

of the reconstructions (i.e. the Frobenius norm of the difference between the true and reconstructed expected adjacency matrices, divided by the norm of the true adjacency matrix) are $0.031$ and $0.894$, respectively; for a direct comparison, with a rank 5 embedding, the error of TSVD is $0.966$.

**Exact Factorization of Real Networks.** In Table 2 we report the exact factorization dimension (EFD) for 11 real-world networks, the rank at which LPCA exactly recovers the network within 2000 iterations (i.e., returns $X, Y$ with $\sigma(XY^T) = A$.). We only compute factorizations at ranks which are multiples of 16, and thus the EFDs are calculated up to a multiple of 16. The values for EFDs are remarkably low – for 3 of the tested networks we achieve exact factorization even at our minimum attempted rank of 16; for these networks, we attempted rank 8 LPCA, but did not achieve exact factorization within 2000 iterations. Moreover, for the rank that we achieved perfect reconstruction (EFD), we report the relative Frobenius error of the TSVD approach; we observe the error is quite high in all cases. For all networks, the EFD is significantly lower than the upper bound presented in Theorem 6, of twice the max degree plus one. With the exception of Pubmed, all network are factored exactly at or below ranks that are twice just the $95^{th}$ percentile degree; the max degree for Pubmed is 171, so it, too, is factored within the theoretical bound.

As a baseline, we generate for each network a set of random graphs with the same expected degree sequence using the algorithm of [VDH16]. We report the EFD at which three random networks generated can be perfectly reconstructed; we note that for a lower rank than the one reported, it was not possible to reconstruct the networks in any of the runs. In general, results are well concentrated and show that EFD is consistently higher for the random networks. In other words, the embeddings capture structure inherent to real-world networks outside just the degree sequence. Understanding this structure more precisely is an interesting direction for future work. For completeness, we repeat the previous experiment generating Erdős-Rényi random graphs with the same expected number of edges. We observe that reconstructing these networks is in fact easier, due to the absence of high degree nodes. This justifies the choice of random networks with the same expected degree sequence as the true networks as a more suitable baseline.

Table 2: Real world graphs for which we find exact adjacency matrix factorizations of the form $A = \sigma(XY^T)$ where $X, Y \in \mathbb{R}^{n \times k}$ and $\sigma(x) = \max(0, \min(1, x))$ is a thresholding function applied entrywise to $XY^T$. EFD is the exact factorization dimension for LPCA. We report the $95^{th}$ percentile degree as a more robust and informative alternative to the maximum degree. TSVD Error is the relative Frobenius error of TSVD at the EFD for LPCA. The final two columns give the EFDs of the random graphs related to these networks described above.

| Data Set | # Nodes | Mean Degree | $95^{th}\%$ Degree | **EFD** | TSVD Error | **EFD** (Exp. Degree) | **EFD** (Erdős–Rényi) |
|---|---|---|---|---|---|---|---|
| Pubmed | 19 581 | 4.48 | 18 | 48 | 0.95 | 48 | 32 |
| ca-HepPh | 11 204 | 21.0 | 90 | 32 | 0.63 | 96 | 64 |
| p2p-Gnutella04 | 10 876 | 3.68 | 32 | 32 | 0.97 | 32 | 16 |
| BlogCatalog | 10 312 | 64.8 | 239 | 128 | 0.71 | 160 | 128 |
| Wiki-Vote | 7 115 | 14.6 | 75 | 48 | 0.77 | 80 | 48 |
| ca-GrQc | 5 242 | 5.53 | 20 | 16 | 0.85 | 32 | 32 |
| Wikipedia | 4 777 | 38.7 | 99 | 64 | 0.69 | 80 | 80 |
| Facebook | 4 039 | 43.7 | 153 | 32 | 0.66 | 96 | 80 |
| PPI | 3 890 | 19.7 | 72 | 48 | 0.81 | 64 | 48 |
| Citeseer | 3 327 | 2.74 | 8 | 16 | 0.94 | 16 | 16 |
| Cora | 2 708 | 3.90 | 9 | 16 | 0.93 | 16 | 16 |

**Recovery of Degree and Triangle Count Sequences.** We next assess the accuracy of very low-dimensional embeddings with respect to reconstructing fundamental network information, namely the sequence of (i) degrees and (ii) participating triangles per node. See our detailed findings in Figures 2 and 3. We see that the LPCA based embeddings are able to capture both sequences near exactly, even with rank much smaller than the EFD. As we range the rank, LPCA's reconstruction quality for both sequences is a monotone function of the rank; TSVD's performance is not monotone as can be seen e.g., in the PPI plots in Figures 2 and 3 for ranks 32 and 128 respectively. Generally, the TSVD method performs significantly worse than LPCA.

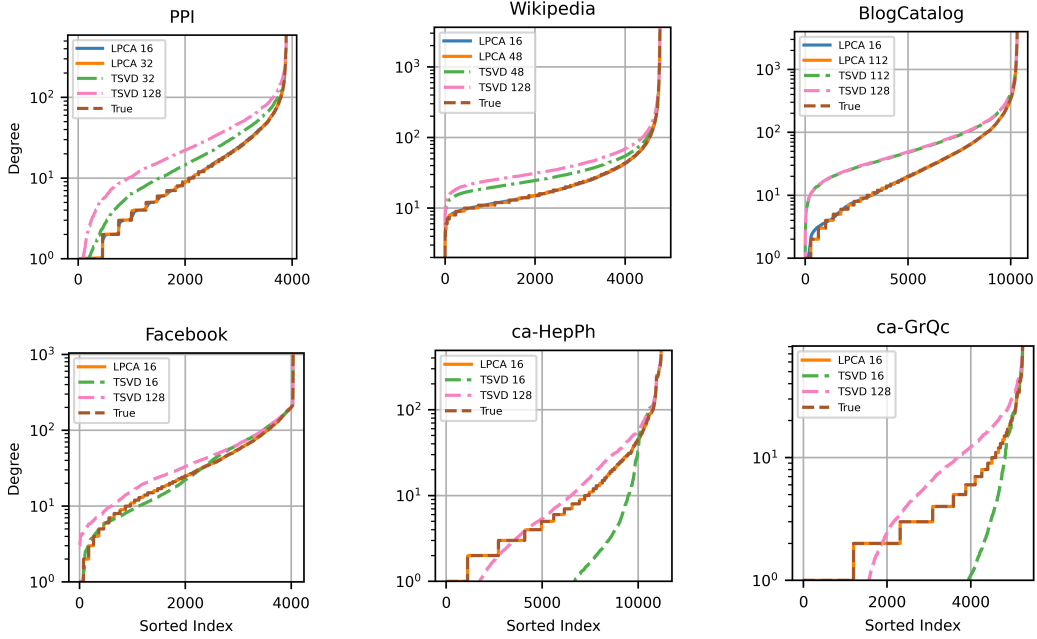

Figure 2: Sorted expected degrees of reconstructed networks.

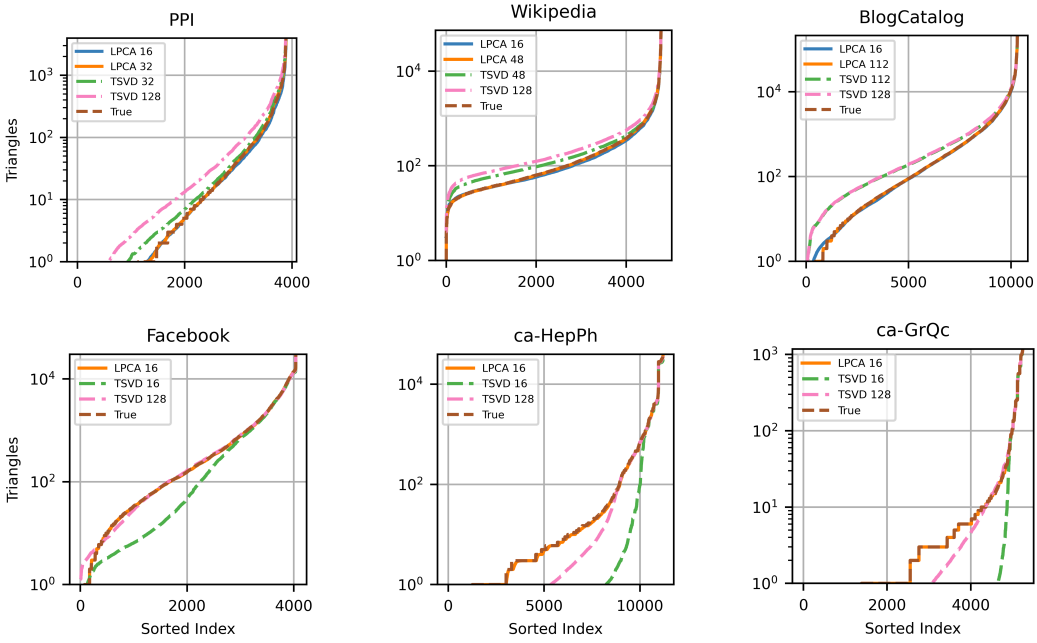

Figure 3: Sorted expected count of triangles involving each vertex in reconstructed networks.

**Recovery of Low-Degree Triangles.** We next turn to the challenge of reconstructing triangles on low-degree nodes, which was the focus of [SSSG20]. We assess the recovery of low-degree triangles in real networks when the embedding rank is not sufficient for exact factorization. Our results are shown in Figure 4 for six networks. The results are representative of what we observe across all our experiments. In each figure, we plot using different factorization ranks the reconstructed normalized number of triangles ($y$-axis) among all nodes whose degree does not exceed a specific upper bound ($x$-axis). We normalize the counts by the number of nodes $n$ to have a consistent measure across all six networks. Notice that the minimum possible non-zero value is $\frac{1}{n}$ and corresponds to exactly one

triangle. We plot the performance of LPCA using rank 16, and for TSVD using rank 128. We also plot the performance of both methods for rank equal to the EFD minus 16. Note that, for `ca-GrQc`, which is reconstructed exactly at our minimum rank of 16, we simply plot rank 16 itself. In addition to the reconstruction results, we also plot the true normalized triangle counts.

In agreement with [SSSG20], we find that the TSVD method consistently underestimates low-degree triangles: whereas the true network begins producing triangles with fairly low-degree vertices, TSVD requires much higher-degree vertices to recover a single expected triangle. This holds at both ranks across the surveyed networks. By contrast, across all of these networks, the just-below-exact rank LPCA tightly matches the true triangle-degree curve. With the exception of `BlogCatalog`, even rank 16 LPCA closely matches the true curve, especially at low degrees. Interestingly, in `BlogCatalog`, while rank 16 LPCA has a higher reconstruction Frobenius error (.82) than either rank 112 TSVD (.73) or rank 128 TSVD (.71), the former still achieves a single expected triangle with lower-degree vertices and overall matches the true triangle-degree curve more closely than the TSVD methods. This seems to suggest an implicit bias of the LPCA method towards capturing local structure in real-world graphs even when factorization is inexact. Understanding this bias more precisely would be an interesting direction for future work. Overall we confirm that LPCA not only outperforms TSVD, but more importantly, illustrates that embeddings can capture the triangle-rich structure of real networks with remarkably accuracy, even at very low ranks where exact factorization is impossible.

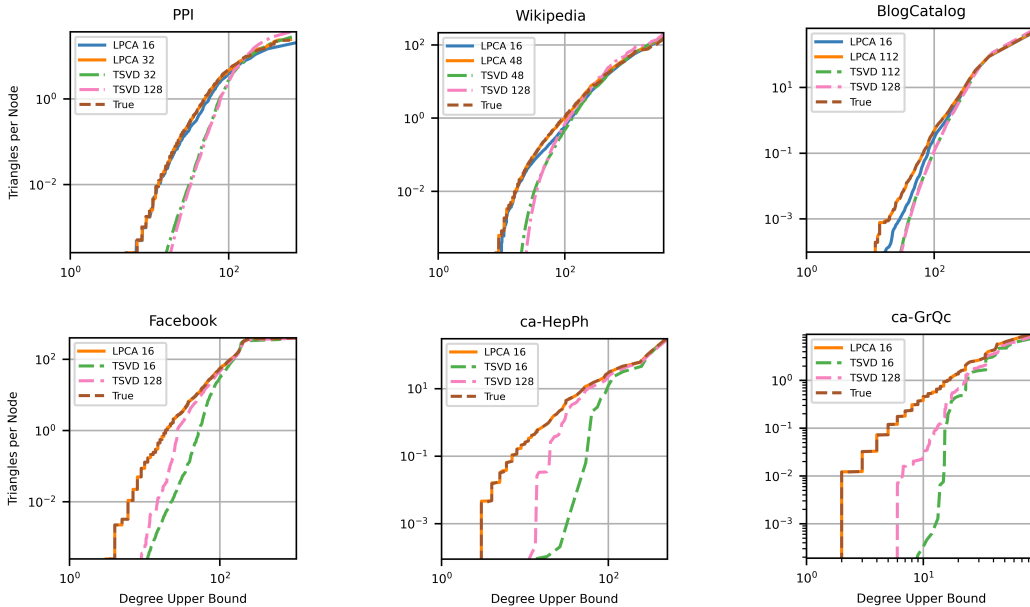

Figure 4: True and recovered counts of triangles in subgraphs induced by nodes whose degree is upper bounded by $c$ ($y$-axis) vs. the degree upper bound $c$ ($x$-axis) for six networks. The recovered triangles have been counted in reconstructed networks using TSVD and LPCA for different ranks.

## 5   Open Problems

Our work leaves several open questions. A key question is if we can strengthen our theoretical results, and better explain the empirical performance of our algorithm on real-world graphs. Answering this would help us understand the type of structure that our embeddings, and perhaps modern node embeddings more broadly, leverage to compress complex networks. From a practical perspective, understanding the connection between the ability of an embedding to reconstruct a graph and performance in downstream classification tasks is an important related question, key to work on graph auto-encoders and the privacy of node embeddings. In initial experiments, we find that our LPCA embeddings do not give good performance in downstream classification tasks. Are there embeddings that simultaneously yield exact or near exact factorizations and good performance in downstream applications? Generally, understanding the strengths and limitations of modern node embedding methods is a broad interesting direction. Our work tackles this question using the perspective of graph factorization, which is just one line of a broader investigation.

## Broader Impacts

This paper contributes towards a better understanding of low-rank factorization, node embedding methods, and shows that a simple algorithm based on logistic PCA can output remarkably accurate low-rank factorizations. Our work may benefit researchers and also practitioners who (i) use node embeddings for downstream machine learning and data analysis tasks, (ii) develop novel node embedding methods, and/or (iii) study the interplay between privacy and node embeddings. Our work illustrates that node embeddings can capture significant specific information about edges in a graph, highlighting potential privacy risks of e.g., releasing such embeddings for users in a social network.

We do not foresee direct negative outcomes of our work, although acknowledge that improved methods for network analysis (including those based on embeddings) have social consequences. For example, improved graph-based recommendations can contribute to issues of filter bubbles, polarization, and the spread of false information [Par11, LCKK14, MMT18]. They may also be associated with negative well-being impacts due to increased social media use [VLP+15, SC17, VYR+17]

## Funding Disclosure

We have no external funding to disclose.

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
