[Supplementary Material]

# A  Deferred Proofs

For completeness, we give a full proof of Theorem 6, which shows that any bounded degree graph admits an exact low-rank factorization. Our proof closely follows the approach of [AFR85] for bounding the sign rank of sparse matrices

**Theorem 6.** *Let $A \in \{0,1\}^{n \times n}$ be the adjacency matrix of a graph $G$ with maximum degree $c$. Then there exist embeddings $X, Y \in \mathbb{R}^{n \times (2c+1)}$ such that $A = \sigma(XY^T)$ where $\sigma(x) = \max(0, \min(1, x))$ is applied entry-wise to $XY^T$.*

*Proof.* Let $V \in \mathbb{R}^{n \times (2c+1)}$ be the Vandermonde matrix with $V_{t,j} = t^{j-1}$. For any $x \in \mathbb{R}^{2c+1}$, $[Vx](t) = \sum_{j=1}^{2c+1} x(j) \cdot t^{j-1}$. That is: $Vx \in \mathbb{R}^n$ is a degree $2c$ polynomial evaluated at the integers $t = 1, \ldots, n$.

Let $a_i$ be the $i^{th}$ row of $A$. $a_i$ has at most $c$ nonzeros since $G$ has maximum degree $c$. We seek to find $x_i$ so that $s(Vx_i) = a_i$, and thus, letting $X \in \mathbb{R}^{n \times (2c+1)}$ have $x_i$ as its $i^{th}$ row, will have $A = s(VX^T)$. This yields the theorem since, if we scale $VX^T$ by a large enough constant (which does not change its rank), all its positive entries will be larger than 1 and thus we will have $\sigma(VX^T) = A$.

To give $x_i$ with $s(Vx_i) = a_i$, we equivalently must find a degree $2c$ polynomial which is positive at all integers $t$ with $a_i(t) = 1$ and negative at all $t$ with $a_i(t) = 0$. Let $t_1, t_2, \ldots, t_c$ denote the indices where $a_i$ is 1. Let $r_{i,L}$ and $r_{i,U}$ be any values with $t_{i-1} < r_{i,L} < t_i$ and $t_i < r_{i,U} < t_{i+1}$. If we chose the polynomial with roots at each $r_{i,L}$ and $r_{i,U}$, it will have $2c$ roots and so degree $2c$. Further, this polynomial will switch signs just at each root $r_{i,L}$ and $r_{i,U}$. We can observe then that the polynomial will have the same sign at $t_1, t_2, \ldots, t_c$ (either positive or negative). Flipping the sign to be positive, we have the result. □

We next give an extension of Theorem 5, showing that a simple binary embedding can yield a graph with very high triangle density.

**Theorem 8** (Simplified Embeddings Capturing Triangles). *Let $\bar{A} = \sigma(UMU^T)$ where $\sigma = \max(0, \min(1, x))$. For any $c$, there are matrices $U \in \{0,1\}^{n \times k}$ and $M \in \mathbb{R}^{k \times k}$ for $k = O(\log n)$ such that if a graph $G$ is generated by adding edge $(i, j)$ independently with probability $A_{i,j}$: 1) $G$ has maximum degree $c$ and 2) $G$ contains $\Omega(c^2 n)$ triangles.*

*Proof.* Let $k = d \log n$ for a sufficiently large constant $d$ and consider binary $U \in \{0,1\}^{n \times k}$ where each row has exactly $2 \log n$ nonzero entries. Let $D = UU^T - \log n \cdot J$ where $J$ is the all ones matrix. Note that $D$ can be written as $UMU^T$ for $M = I - \frac{1}{4 \log n} J$.

Observe that the only positive entries in $D$ are those where $u_i^T u_j > \log n$. Thus $\bar{A} = \sigma(D)$ is binary with 1s where $u_i^T u_j > \log n$ and 0s elsewhere. In turn, $G$ is deterministic, with adjacency matrix $\bar{A}$.

We will construct $U$ so that its rows are partitioned into $n/c$ clusters with $c$ nodes in them each as in Theorem 5. The construction is as follows: choose $n/c$ random binary vectors $m_1, \ldots, m_{n/c}$ (the 'cluster centers') with exactly $2 \log n$ nonzeros in them. In expectation, the number of overlapping entries between any two of these vectors will be $\frac{2 \log n}{d}$ and so with high probability after union bounding over $\binom{n/c}{2} < n^2$ pairs, all will have at most $\frac{\log n}{3}$ overlapping entries if we set $d$ large enough. Thus, $m_i^T m_j < \frac{\log n}{3}$ for any $i$ and $j$ and the centers will not be connected in $G$.

If we set $d$ large enough, then around each cluster center $m_i$, there are at least $\binom{d \log n - 2 \log n}{\log n / 3} \geq n \geq c$ binary vectors $v_1, \ldots, v_c$ each with $2 \log n$ nonzeros that overlap the center on all but $\frac{\log n}{3}$ bits and so have $m_i^T v_j > 2 \log n - \frac{\log n}{3} > \log n$ and a connection in the graph.

Additionally, each $v_i$ must overlap each other $v_j$ in the same cluster on all but at most $\frac{2 \log n}{3}$ bits and so $v_i^T v_j \geq 2 \log n - \frac{2 \log n}{3} > \log n$ and so they will be connected in the graph. Finally, each $v_i$ overlaps each center of a different cluster on at most $\frac{2 \log n}{3} < \log n$ bits, and so there are no connections between clusters. So $G$ is a union of $n/3$ sized $3$ cliques, and so by the same argument as Theorem 5 has maximum degree $c - 1$ and $\Omega(c^2 n)$ triangles, giving the theorem. □