[Reviews · NeurIPS 2020]

Review 1

Summary and Contributions: The paper deals with the problem of learning node embeddings based on low-rank matrix factorization. The main question asked concerns the extent to which low-dimensional embeddings can reconstruct sparse graphs with high triangle density – both key properties that can be found in various real-world graphs. In particular, the paper builds upon recent work, which gives a negative answer to the above question. The authors prove that under a non-PSD factorization of the adjacency matrix, the reconstructed graph produced by a simple thresholding function will have a specific maximum node degree and number of triangles. In other words, low-rank factorization models can simultaneously lead to sparsity and high triangle density. Besides, the paper proves exact low-dimensional embedding results for bounded degree graphs. Lastly, by properly choosing the reconstruction loss functions, a new node embedding model is proposed. In almost all cases, the authors theoretically support the findings of the paper. Lastly, empirical results on real-world graphs support the theoretical analysis of the paper.

Strengths: - The paper deals with an important class of models for learning node embeddings, improving our understanding of low-rank factorization models. - A thorough study of the embeddings of bounded degree graphs as well as analysis of the preferential-attachment model. - The paper introduces a simple algorithm that can significantly improve the quality of the representations.

Weaknesses: - The analysis is performed on low-rank factorizations of the adjacency matrix of the graph. Although very interesting, this is quite limited; the majority of recent embedding models factorize more ‘informative’ node proximity matrices. - Similar to the above, the empirical analysis is restricted to simple low-rank factorizations of the adjacency matrix. I guess, considering powers of the adjacency matrix, it will impact some of the proposed properties (e.g., sparsity). - Although not directly in the scope of the paper, it would be interested to examine the impact of the embeddings on tasks beyond graph reconstruction (e.g., link prediction which is very similar, as well as node classification).

Correctness: The claims of the paper seem correct to me. I have tried to examine the proofs and couldn't find any flaw. The empirical methodology also seems correct to me. As I mentioned above though, I would expect to have a more elaborate empirical evaluation including baselines that leverage proximity matrices beyond the adjacency.

Clarity: The paper is well-written. Most of the arguments made are supported theoretically.

Relation to Prior Work: The paper contains detailed description of the prior work. What I missed though is the fact that prior methods have not been used in the empirical evaluation (please see more comments on that below).

Reproducibility: Yes

Additional Feedback: The main findings of the paper are interesting and supplement/extend recent work on the topic. Besides, the proposed reconstruction algorithm, although quite simple, provides interesting results. The are some point though that are not clearly presented or addressed in the paper, and some further investigation might be needed. Most of these points have been mentioned above. Here I will elaborate in more detail: - The analysis is performed on low-rank factorizations of the adjacency matrix of the graph. Although this offers a simple framework upon which interesting theoretical results can be obtained, most of the recent matrix factorization embedding models rely on higher-order proximity matrices. How are the tools developed in the paper able to deal with such models? Is there any generalization? - Similarly, for the empirical analysis. To have a broader view of the capabilities of the proposed model, it would be interesting to examine how higher-order models behave. - In Theorem 5, how arbitrarily large c (maximum degree) can be? In order to control the sparsity of the graph, shouldn’t we focus on the expected degree instead of c? - In Theorem 6, I missed the connection made before to graphs of low sign-rank. For the argument made here, is \sigma() used as shown in the paper or the sign function s()? - What I also missed from the paper, is a more generic evaluation framework, including link prediction and node classification. Even though the paper focuses on graph reconstruction, it would be very interesting to have some discussion/experiments on link prediction (for which the results are expected to be quite similar) and node classification. Typos: - Line157: ‘… mor*e* …’ ==================== I would like to thank the authors for their response. After reading the rebuttal and the comments of the rest reviewers, I decided to retain my score. ====================


Review 2

Summary and Contributions: In this paper, the authors first shown that limitation in the previous impossibility theorem, and prove that graphs with bounded degree and abundance of triangles can be captured by a low-rank model. Furthermore, the authors propose an algorithm that can produce the explicit low-rank decomposition. The experiments demonstrate that the proposed LPCA algorithm is significantly better than the previous TSVD algorithm, and strongly support the theoretical results.

Strengths: Soundness: The paper has strong theoretical results and validates those with a good selection of experiments. Significance and novelty: Both the theoretical results and the algorithm are very important contributions. The former shows the limitation in the impossibility theorem by Seshadhri et al., and proposes an alternative formulation with proof of existence. The LPCA algorithm is a significant improvement over the previous TSVD algorithm. It constructs a low-rank factorization that is often even lower than the theoretical bound. Node embedding has attracted much attention in the recent years, and I think the results of this paper make very valuable contribution to this problem. I think the work is novel. Relevance: The results in this paper is highly relevant for the audience of this conference.

Weaknesses: I do not think the paper has any outstanding weakness. Although not the focus of this paper, I would like see the embedding used in some popular applications.

Correctness: I believe the claims and method is this paper are correct. They are also well supported by the empirical evidence.

Clarity: The paper is very well-written and easy to follow. A little more details can be included, maybe in the supplementary section. For example, I'm not familiar with the result in theorem 6, so it would be great for authors to include a sketch or point out the reference here. My other confusion is that Result 2 seems to suggest the optimization for LPCA has guaranteed global convergence to exact A. Could the authors confirm if that's the case and give a bit more detail?

Relation to Prior Work: The relation to prior work is stated clearly. The paper contains a good and complete list of references.

Reproducibility: Yes

Additional Feedback: 1. This paper exclusively focus on the relation between degree distribution and triangle counts. Does LPCA also has superior performance in terms of other substructures or proximity metrics? 2. I would like to see more details about the optimization of for LPCA. Is global convergence guaranteed? I think scalability could be problem for the algorithm because XY^T need to be explicitly formed over a large number of iterations. In that case what are the limit of LPCA in terms of graph size? Finally, is it possible to avoid the issue that rank k has to be determined a priori? Could we use the theoretical bound as the starting k, and add a regularization term to the optimization problem, maybe L_{infty, 1}, to encourage minimal rank? 3. Could the authors comment on the implication of this work on popular applications (e.g. node clustering/ link prediction, etc)? Given that the new model captures the abundance of triangles much better than previous factorization-based method, will it also have better performance in those tasks? On the other hand, how does it compare to other node embedding techniques (e.g. NN-based)?


Review 3

Summary and Contributions: The paper focuses on the relation between the low-dimensional embeddings of node and the basic properties of the real-world graph, i.e, both the sparse and high triangle density. It proves that a minor relaxation of the model used in [SSG20] can generate sparse graphs with high triangle density, also this model leads to exact low-dimensional factorizations of many real-world networks. The paper gives the LPCA algorithm to find such exact low-embeddings. The author conduct experiments on synthetic and real-worlds networks to verify the ability of very low-dimensional embeddings to capture local structure of the graphs.

Strengths: 1. The paper show the results of Seshadhri are not related to the low-dimensional structure of complex networks. 2. Prove that a minor relaxation of their model can generate sparse graphs with high triangle density. 3. Provide sufficient experimental evaluation and achieve promising empirical result.

Weaknesses: 1. The analysis and interpretation for the design of LPCA is limited, the exact param-settings are not given explicitly. 2. The approach proposed in the paper, i.e. the minor relaxation for the model in [SSSG20] by introducing two embedding X and Y for the node embedding, may be not exactly the original problem in [SSSG20], which consider the PSD low-rank embedding for the graph; 3. There is not comparison with some SOTA network embedding methods, and the generalization of the conclusion for other node-embedding methods. The datasets is relatively small. Whether the algorithm is effective on such a large-scale network and whether the generated network still has the nature of a real network are issues that need to be verified.

Correctness: may be correct. [but not check all the details of proof.]

Clarity: It is basically well written, but also contains some typos. E.g. 1. ‘mor’ in Line 157; 2. ‘As discussed in Section 5, many natural …’ in Line 212, there is Sec 5. The poofs makes it hard to follow.

Relation to Prior Work: The author discusses the relation between this work and previous work, and analyses the limitation of prior work, highlights the contribution of this work. Also, the main motivation of the paper is inspired from the prior work [SSSG20]. Some node embedding works based on deep neural networks are missing.

Reproducibility: Yes

Additional Feedback: The paper focuses on the relation between the low-dimensional embeddings of node and the basic properties of the real-world graph, i.e, both the sparse and high triangle density. It proves that a minor relaxation of the model used in [SSG20] can generate sparse graphs with high triangle density, also this model leads to exact low-dimensional factorizations of many real-world networks. The paper gives the LPCA algorithm to find such exact low-embeddings. The author conduct experiments on synthetic and real-worlds networks to verify the ability of very low-dimensional embeddings to capture local structure of the graphs. The motivation is clear, it partially answers the problem for the limitation of low-dimensional embedding by the graph factorization. The proposed LPCA algorithm is simple and clear, but the analysis and interpretation for the design of LPCA is limited, the exact parameter settings are not given explicitly. Also, the author provides sufficient experimental evaluation and LPCA achieves promising empirical result. However, I doubt whether it is still the original concern problem in [SSSG20] by introducing the ‘the minor relaxation for their model’, the proposed approach considers two embedding X and Y for the node embedding and it also not satisfies the constraints of PSD low-rank embedding for the graph in [SSSG20]. But we also appreciate the promising empirical result achieved by LPCA compared with baselines. Moreover, there is not comparison with some SOTA network embedding methods, and the generalization of the conclusion for other node-embedding methods, which maybe also our common interest.


Review 4

Summary and Contributions: - Showed theoretically and experimentally that representing each node as two vectors solves a fundamental limitation (i.e., underestimating the number of triangles with low-degree nodes) of node embedding where each node is represented as a single vector (Theorem 5). - Proposed a non-linear factorization model for adjacency matrices and provided an upper bound of the rank for exact recovery (Theorem 6). - Proposed an optimization algorithm (LCPA) for the factorization model. LCPA requires lower ranks for exact recovery than truncated SVD (TSVD).

Strengths: - The claims are well supported both theoretically and empirically. - The constructive proof of Theorem 5 is based on a non-trivial idea. - The LCPA is evaluated thoroughly using multiple metrics on 11 real-world graphs.

Weaknesses: - The usefulness of LPCA is not clear. It could be applied to down-stream tasks such as link prediction and clustering. - Only TSVD is used as a baseline approach. More embedding methods (e.g., DeepWalk and Node2Vec) could be considered as baseline. (After rebuttal: I found that more baselines were analyzed well in Seshadhri et al. Please clarify this in the final version. I adjusted my score based on this). - Theorem 6 presents an important result but the proof seems to be missing (not found in the supplement).

Correctness: I could not find any flaw in the proofs and empirical methodologies except for that the baseline embedding method, which the authors call Truncated SVD (TSVD), seems different from conventional TSVD.

Clarity: Yes. The claims on the contributions are made clear and supported with either proofs or experiments. The proofs are understandable. Experimental results are clear in demonstrating the advantages of the proposed method.

Relation to Prior Work: Yes. The authors claimed that the limitation pointed out in the previous work (SSSG20) is the limitation of the proposed model in that paper. They supported their claim by the proof of Theorem 5 and the experimental results of their methods on various criteria.

Reproducibility: Yes

Additional Feedback: - In the last paragraph of page 6, it is hard to understand why 'EFD is consistently higher for the random networks" is equivalent to "the embeddings capture structure inherent to real-world networks outside just the degree sequence". Perhaps some more detailed explanation here would have readers understand the authors' point better.

[Author Response · NeurIPS 2020]

We thank the reviewers for their detailed reviews and suggestions. We address the most relevant concerns below.

**Performance of LPCA embedding in downstream tasks (e.g., link predication and node clustering):** While
beyond the scope of our current work, the performance of our embeddings in downstream tasks is a very interesting
question. In our work we show affirmatively that low-dimensional embeddings can in fact capture structural information
like the triangle density and degree sequence. A broad next question we seek to answer is: how does the ability to
preserve such structural information impact an embedding's usefulness in downstream application?

**Comparison to baselines beyond TSVD (e.g., DeepWalk, Node2Vec):** The prior work for Seshadhri et al. gives a
detailed comparison between node2vec and TSVD, which are comparable on triangle density reconstruction. We will
clarify this in the final version. It would be interesting to expand this comparison to other methods, although, related to
the question above, such methods may perform worse than LPCA on graph construction but still perform well on other
downstream tasks. We are actively working on follow up work which seeks to 'invert' modern embeddings to reveal
graph structure such as triangle density, the degree sequence, and edges.

**Reviewer 1:**

*The analysis is performed on low-rank factorizations of the adjacency matrix of the graph. Although very interesting,*
*this is quite limited; the majority of recent embedding models factorize more 'informative' node proximity matrices.*

As the reviewer suggests, capturing higher order connectivity relates closely to today's most popular embedding
methods. We view the submission as a step in this direction. Before our work, even for the simple adjacency matrix, the
power of low-dimensional embeddings was previously unclear, and the large gap between using one embedding per
node (a PSD embedding as in Seshadhri et al.) vs. two (a non-PSD embedding as in our work) was unknown.

*In Theorem 5, how arbitrarily large c (maximum degree) can be?...shouldn't we focus on the expected degree instead?*

Theorem 5 holds for any $c$, from 1 to $n - 1$. The expected degree of our construction is the same as the maximum
degree (the graph is regular), so the same theorem holds with 'maximum degree' replaced by 'expected degree'.

*In Theorem 6, I missed the connection... to sign-rank... is $\sigma()$ used as shown in the paper or the sign function $s()$?*

$\sigma()$ is used as shown. If we scale the entries of $X, Y$ by a large enough value then $\sigma(XY^T)$ exactly equals $s(XY^T)$,
so any results for sign rank directly apply in our model. We will expand the discussion of this in the final version.

**Reviewer 2:**

*Result 2 seems to suggest the optimization for LPCA has guaranteed global convergence...Could the authors confirm...*

The low-rank LPCA objective is non-convex and convergence to a global minimum is not *guaranteed*. However, for
all graphs tested, we achieved exact recovery and so did in fact converge to a global minimum. Understanding this
theoretically would be very interesting, as would be applying relaxation methods as suggested by the reviewer, which
may give provable convergence and/or not require setting the rank $k$ a priori.

*Does LPCA have superior performance in terms of other substructures or proximity metrics? (outside degree/triangles)*

We also performed experiments on 5-cycle density and other graph motifs, showing superior performance of LPCA. We
will add to the supplemental material. When LPCA gives an exact factorization, all substructures are exactly preserved.

**Reviewer 3:**

*The minor relaxation for the model in [SSSG20] by introducing two embedding X and Y for the node embedding, may*
*be not exactly the original problem in [SSSG20], which consider the PSD low-rank embedding for the graph.*

This is correct. However, we argue that our relaxation is in fact more natural. Many modern embedding methods do not
produce a PSD factorization, and thus fall into our model, but not the one of [SSSG20].

**Reviewer 4:**

*In the last paragraph of page 6, it is hard to understand why 'EFD is consistently higher for the random networks' is*
*equivalent to "the embeddings capture structure inherent to real-world networks outside just the degree sequence'.*

Thanks – we will clarify the discussion. At a high level, higher EFD for the random networks suggests that real-world
networks must have some structure that can be compressed, leading to lower EFD. More explicitly understanding this is
a key direction for future work, related to understanding gaps between real world networks and random graph models.

*Theorem 6 presents an important result but the proof seems to be missing (not found in the supplement).*

We will give a detailed proof in the final version. See our answer to Reviewer 1 above which explains the connection to
sign rank and gives a proof sketch.

[Meta-Review · NeurIPS 2020]

The referees found this to be a very well-written paper with interesting and valuable insights (both methodological and algorithmic) for low-rank embeddings in graphs. The proposed low-rank embedding function is also simple to interpret and opens the door for future research. The only outstanding question is how well these types of embeddings perform on downstream tasks not involving graph reconstruction, but the authors appropriately discuss this as an open direction. This paper should be accepted.